# Determining Frequency of Common Pulmonary Gross and Histopathological Findings in Feedyard Fatalities

**DOI:** 10.3390/vetsci10030228

**Published:** 2023-03-16

**Authors:** Paige H. Schmidt, Brad J. White, Abigail Finley, Eduarda M. Bortoluzzi, Brandon E. Depenbusch, Maddie Mancke, Rachel E. Brown, Makenna Jensen, Phillip A. Lancaster, Robert L. Larson

**Affiliations:** 1Beef Cattle Institute, Kansas State University, Manhattan, KS 66506, USA; 2Department of Veterinary Pathobiology, Texas A & M School of Veterinary Medicine, College Station, TX 77843, USA; 3Irsik and Doll Feed Services, Inc., Cimarron, KS 67835, USA

**Keywords:** necropsy, bovine respiratory disease, feedyard, histopathology, lung, bronchopneumonia, acute interstitial pneumonia

## Abstract

**Simple Summary:**

Respiratory disease continues to play a major role in deaths seen in feedlot cattle. The most common respiratory diseases include bronchopneumonia, acute interstitial pneumonia, and bronchopneumonia with an interstitial pneumonia. Necropsy is a useful tool to collect information and enhance decision making about the health and management of a pen or lot. A more refined diagnosis and estimation of pulmonary lesions in all cattle, regardless of clinical signs, can provide a more accurate estimate of respiratory disease frequency. A diagnosis was determined during necropsy and further diagnostics were submitted for histopathologic evaluation to determine the agreement between the two diagnostics modalities. Both methods reported similar frequencies between the three most common respiratory diseases. The most frequent respiratory diseases recognized were bronchopneumonia and bronchopneumonia with an interstitial pneumonia. Recognizing bronchopneumonia with an interstitial pneumonia from bronchopneumonia and acute interstitial pneumonia can be an important component to understanding these disease processes and the cattle affected.

**Abstract:**

Pulmonary disease is often associated with feedlot cattle mortality, and the most common syndromes include bronchopneumonia, acute interstitial pneumonia, and bronchopneumonia with an interstitial pneumonia. The study objective was to utilize gross necropsy and histopathology to determine the frequency of pulmonary lesions from three major syndromes and agreement between gross and histopathological diagnosis. A cross sectional, observational study was performed at six U.S. feedyards using a full systematic necropsy to assess mortalities during summer 2022. A subset of mortalities had four lung samples submitted for histopathological diagnosis. Gross necropsy was performed on 417 mortalities, 402 received a gross diagnosis and 189 had a histopathological diagnosis. Descriptive statistics were used to evaluate pulmonary diagnosis frequency based on method (gross/histopathology), and generalized linear mixed models were used to evaluate agreement between histopathological and gross diagnoses. Using gross diagnosis, bronchopneumonia represented 36.6% of cases with acute interstitial pneumonia and bronchopneumonia with an interstitial pneumonia representing 10.0% and 35.8%, respectively. Results identified bronchopneumonia with an interstitial pneumonia as a frequent syndrome which has only been recently reported. Histopathological diagnosis had similar findings; bronchopneumonia represented 32.3% of cases, with acute interstitial pneumonia and bronchopneumonia with an interstitial pneumonia representing 12.2% and 36.0%, respectively. Histopathological diagnosis tended (*p*-VALUE = 0.06) to be associated with gross diagnosis. Pulmonary disease was common and both diagnostic modalities illustrated three primary syndromes: bronchopneumonia, acute interstitial pneumonia, and bronchopneumonia with an interstitial pneumonia with similar frequencies. Improved understanding of pulmonary pathology can be valuable for evaluating and adjusting therapeutic interventions.

## 1. Introduction

Pulmonary disease is a major issue in feedyard cattle resulting in significant performance and economic impacts [1,2,3,4]. A variety of etiologic and risk factors contribute to pulmonary disease and an accurate diagnosis impacts effectiveness of prevention and control programs. Necropsy is a useful diagnostic tool to collect information in order to enhance decision making about the health and management of a pen or lot. Respiratory disease contributes to feedyard cattle mortalities throughout the feeding phase [5,6]. Several previous studies have evaluated the timing, infectious components, and economic loss of respiratory disease in feedyard cattle [7,8]; however, limited research has compared gross and histopathological lesions and the linkage to cattle demographics. Improved diagnostics in postmortem cattle will increase knowledge of disease processes, timing, and demographics of affected cattle, ultimately leading to more precise therapeutic interventions and efficiency.

Previous studies evaluated gross and histopathological lesions in feedyard cattle at death; moreover, these studies are often limited to cattle with confirmed clinical respiratory disease prior to death [9,10]. Cause of death is often recorded as a singular disease which may restrict the true estimation of pulmonary lesions when concurrent disease processes are present. Estimating pulmonary lesions in all cattle regardless of clinical diagnosis or concurrent pathological lesions can provide a more accurate estimate of pulmonary disease frequency. Improved classification regarding cause of death seen throughout the feeding phase will allow a clearer understanding of these disease processes and the cattle they affect. Pulmonary disease has been documented in cattle at harvest with no previous signs of disease [11,12], but little work has been done documenting pulmonary disease in cattle that died of other causes.

A major syndrome associated with pulmonary disease in feedlot cattle is bronchopneumonia (BP) which plays a significant role in morbidity and mortality seen throughout the feeding period. Pathological agents associated with BP include viruses such as Bovine herpesvirus 1 (BHV-1), Bovine parainfluenza virus (BPIV-3), Bovine viral diarrhea virus 1 and 2 (BVDV 1, 2), Bovine respiratory syncytial virus (BRSV), Bovine adenovirus A-D (BAdV A, DP and Bovine coronavirus (BcoV), along with bacterial pathogens *Mannheimia haemolytica*, *Pasteurella multocida, Histophilus somni* and *Mycoplasma* spp [13,14,15,16]. Depending on etiologic agent(s), gross lesions of BP can have various presentations. Acute interstitial pneumonia (AIP) is a sporadic, rapidly progressing respiratory disease, with high mortality in feedyard cattle [17,18,19]. Multiple pathogenic processes have been considered for inciting AIP; however, little is known about risk factors for this disease [20,21]. Bronchopneumonia with an interstitial pneumonia (BIP) is a combination of pulmonary lesions consistent with both BP and AIP [22,23]. While a diagnosis of BIP is recorded by some veterinarians, other diagnosticians will attribute cause of death to either BP or AIP. The risk of BIP compared to lesions associated with only BP or AIP is not well documented. 

The study objective was to utilize gross necropsy to determine the frequency of pulmonary lesions associated with three major diagnoses: acute interstitial pneumonia (AIP), bronchopneumonia (BP), and bronchopneumonia with an interstitial pneumonia (BIP). Furthermore, we assessed the accuracy of recognizing AIP, BP, and BIP gross respiratory lesions to corresponding histopathological samples and if there was consistency between diagnosis throughout the four samples collected from the right cranioventral, left cranioventral, right caudodorsal and left caudodorsal lobes of each lung. 

## 2. Materials and Methods

The Institutional Animal Care and Use Committee (IACUC) was contacted, and a protocol was deemed unnecessary due to evaluations only performed on cattle mortalities. 

### 2.1. Experimental Design

This cross sectional, observational study was designed to determine gross pulmonary lesions observed throughout the feeding period in feedlot cattle mortalities from all causes. Feedlot cattle were necropsied from 1 June 2022 to 29 July 2022, at six feedyards located in the high plains region of the United States. Mortalities enrolled in the study were from deceased cattle within eighteen hours of postmortem examination and/or minimal to no gross autolysis. Autolysis was determined by the individual performing the postmortem examination based on external appearance, tissue color, smell, and texture. Individual calf demographics were provided by the feedyard following the necropsy. 

### 2.2. Postmortem Evaluation

A full systematic necropsy was performed by modifying previously described procedures [24]. All necropsies were documented, and photographs were collected. Necropsies were conducted in groups of two veterinary technicians [PHS, MM, RB, MJ] with one technician performing the postmortem examination and determining the gross lesions. Postmortem diagnoses based on gross findings were confirmed by a veterinarian [BJW]. 

Gross pathological pulmonary lesions were categorized into four areas: BP, AIP, BIP, and Undifferentiated. Diagnosis of BP was based on a variety of characteristics reported in previous research including lung consolidation, interlobular fibrinous material, pulmonary abscesses and firm/rubbery texture of lungs by palpation [10,25,26]. Cattle diagnosed with AIP lesions had diffuse, overinflated lung lobes, interlobular edema and emphysema, and pulmonary lobules varied in color from light pink to dark red leaving a “checkerboard” appearance which has been described in previous research [19,20,27,28]. Lung lesions that expressed characteristics of both bronchopneumonia and an interstitial pneumonia were diagnosed as BIP [16,22,23]. Gross pathological lesions for BIP typically included a diagnosis of BP in the cranioventral lung lobes and a diagnosis of interstitial pneumonia in the caudodorsal regions of the lungs. A line of demarcation was often noted between the two pathological processes within the lungs. Examples of gross lesions consistent with each classification are depicted in Figure 1. Respiratory lesions that did not meet the above criteria (AIP, BP, or BIP) were recorded and documented but categorized as an Undifferentiated pulmonary lesion for the purpose of this study (i.e., Embolic pneumonia). The case inclusion strategy was utilized to create a dataset consisting of all animals necropsied with a gross pulmonary diagnosis (GrossDx) as described in Figure 2.

### 2.3. Histopathology

To maximize study resources, lung samples were collected from the first 9–11 mortalities at each feedyard each week. Regardless of the number of mortalities and postmortem diagnoses, a maximum of 11 cases were taken from one feedyard in a week. Pulmonary histopathology samples (HistoSp) were collected as a 1 × 1 cm square from four areas of the lung: right cranioventral, left cranioventral, right caudodorsal and left caudodorsal lung lobes. If applicable, samples were obtained at the junction between grossly diseased and non-diseased lung tissue. 

Lung samples were fixed in 10% neutral buffered formalin. Individual samples from each case were dyed with one of four different colors of Davidson Tissue Dye for sample location identification. For histopathology processing, standard techniques were utilized to prepare samples. Slides were placed in the Sakura Prisma Plus Stainer, stained with hematoxylin and eosin (H&E), and cover-slipped according to the manufacturer’s protocol. Slides were loaded into the Hamamastu NanoZoomer S360 digital slide scanner and read by a pathologist who was blinded to gross diagnosis determined in the field. 

An individual sample diagnosis was determined using specific histopathological criteria (HistoSp), and each case could have 4 histopathological diagnoses. Microscopically, the lesions of AIP included an alveolar septum expanded by mononuclear inflammatory cells, alveolar septal necrosis, and hyaline membranes. By definition, all AIP/BIP cases had either hyaline membranes and/or type II pneumocyte hyperplasia. Hyaline membrane are characterized by eosinophilic lamellations of polymerized fibrin and necrotic debris that replace lost type I pneumocytes of the alveolar interstitium. Some cases also had type II pneumocyte hyperplasia and bronchiolitis obliterans [9,27,29]. Bronchopneumonia type lungs were categorized by neutrophils within the bronchi/bronchioles and/or alveoli. Figure 3 provides example photomicrographs for each of the diagnoses. For the purpose of this study, histopathological lung samples that did not meet the above criteria were categorized as Undifferentiated. Individual samples were excluded if histopathological autolysis was noted in one or more lung samples or there was inadequate location identification due to absence of tissue dye. 

To allow comparison with the animal-level gross diagnosis, histopathological diagnoses were aggregated to the animal level (HistoDx). Animals were only eligible to be included in the dataset with an animal level histopathological diagnosis if all four lung samples were evaluated from an individual and if none of the samples had autolysis rendering the diagnosis impossible. Each case was classified by combining findings from each of the four lung samples to create a single classification for a case. If the four lung samples were diagnosed with only BP or AIP the case was considered BP or AIP, respectively. All four lung samples did not have to be positive to receive a single diagnosis, but no lesions consistent with AIP were noted in BP cases and vice versa. Cases with both AIP and BP in any combination of the four lung samples were considered to be BIP. In cases where no diagnosis of AIP or BP were attained, samples were considered Undifferentiated. In addition, samples presenting healthy lung tissue were also allocated as Undifferentiated. The sampling strategy resulted in a dataset containing the HistoDx for each animal (Figure 2). Grossly autolyzed cases were excluded at the time of necropsy; however, gross autolysis can be difficult to distinguish from true necrosis. Further autolysis was evaluated during the histopathology examination and cases with autolysis in one or more lung sample were excluded. 

### 2.4. Statistical Analysis

Descriptive statistics were performed on all cases relative to known demographic factors. Two data sets were created to compute the agreement of: (1) GrossDx with HistoDx; (2) HistoSp with HistoDx. A generalized linear mixed effect model was created to evaluate the probability of agreement between HistoDx and GrossDx using the glmer function from the lme4 package of R Studio (https://cran.r-project.org/web/packages/lme4/ (1 October 2022)). Feedyard was included as a random effect to account for the lack of independence among cases. A second generalized mixed effect model was used to evaluate the likelihood that HistoSp was associated with HistoDx using the glmer function from the lme4 package of R Studio. Feedyard and individual animal case number were used as random effects to account for the lack of independence among histopathological samples. The lung sample location and potential interaction with HistoDx were evaluated in the HistoSp model. 

## 3. Results

Four hundred and seventeen necropsies were performed. Thirteen cases were excluded due to gross autolysis. Two cases were excluded because an incomplete necropsy was performed due to carcass removal for rendering prior to completion of necropsy. Four-hundred and two complete necropsies from 6 feedyards met the inclusion criteria. Cattle demographics can be seen in Table 1. Heifers made up 69.4% of the mortalities, while the remaining 30.6% were steers. This was consistent with the feedyard’s live cattle enrollment. Cattle of native origin made up 93% of the mortalities. The remaining cattle included Beef × Dairy, Holstein, or Mexican origin. Arrival weight ranged from 181–453 kg with most mortalities presenting arrival weights between 318 and 363 kg (61.7%). Arrival weights for 5 calves were not obtained due to record errors. Days on feed (DOF) prior to death ranged from 2–222 days. Cattle less than 100 days on feed comprised 55% of the mortalities examined, followed by 30.3% between 101–150 DOF. Days on feed for 3 calves were not obtained due to record errors. 

Gross and histopathological samples were collected each week at 6 different feedyards (Table 2). The sampling strategy at each feedyard was to perform gross necropsies on all eligible cattle at each facility and due to variance in animals present at each facility the number necropsied varied by facility and week.

### 3.1. Gross Results

Of the 402 cattle with gross necropsy results, 359 cases showed gross lung lesions (89%). The remaining forty-three cases were not evaluated due to the absence of a lung lesion. Of the 402 cases that were grossly diagnosed, 36.6% (147/402) were classified with BP, 10.0% (40/402) with AIP and 35.8% (144/402) were classified with BIP (Figure 4). The remainder of the cases were categorized in the Undifferentiated category, including healthy lungs (17.7%; 71/402). 

### 3.2. Histopathological Results

Lung samples were taken from 318 mortalities and submitted for histopathology. Twenty-four samples were lost in transit resulting in 294 cases evaluated by the pathologist. Of the 294 cases, 102 were excluded due to microscopic autolysis in one or more of the four samples collected. An additional three cases were excluded due to inadequate labeling. Thus, histopathological lung samples (HistoSp) from 189 cases were considered for this evaluation. 

Similar to the distribution of GrossDx, BIP lesions accounted for 36.0% (68/189) of the lung HistoDx. BP followed with 32.3% (61/189) and AIP was classified in 12.2% (23/189) of the HistoDx. Respiratory lesions that did not fit the above three categories or healthy lungs were categorized as Undifferentiated (19.6%; 37/189) (Figure 5). 

### 3.3. GrossDx versus HistoDx

The cases where both HistoDx and GrossDx were known (*n* = 172) were evaluated for potential associations between the two diagnoses. The HistoDx tended to be (*p*-VALUE = 0.06) associated with GrossDx. Case classification by each of the two diagnostic methodologies is demonstrated in Table 3. AIP GrossDx had a low probability of agreement with AIP HistoDx (*p*-VALUE = 0.23); many of these cases were classified as BIP with HistoDx (*n* = 10). BIP GrossDx and HistoDx had the highest probability of agreement (*p*-VALUE = 0.49). Gross BIP cases that were not classified as HistoDx BIP were classified as exclusively BP or AIP with HistoDx. BP GrossDx and HistoDx had a 0.36 probability of agreement. The majority of BP GrossDx cases that did not agree were diagnosed as BIP with HistoDx. Finally, the lowest probability of agreement was between the Undifferentiated lung lesions (*p*-VALUE = 0.17). HistoDx classified 18 of the 24 Undifferentiated GrossDx cases as BP. 

### 3.4. HistoDx and HistoSp

A generalized mixed regression model was used to evaluate the probability of diagnostic agreement between each individual HistoSp diagnosis (4 per mortality) with the overall HistoDx (*n* = 189). Sample location was not significantly (*p*-VALUE = 0.11) associated with the probability of agreement between HistoSp and HistoDx. The potential interaction between sample location and HistoDx was also not significantly (*p*-VALUE = 0.22) associated with probability of agreement between HistoSp and HistoDx. In the final model, only HistoDx was significantly (*p* < 0.01) associated with probability of agreement between HistoSp and HistoDx.

The highest probability of agreement between HistoSp and HistoDx was seen for Undifferentiated (probability = 0.99) and AIP (probability = 0.89) (Figure 6). The HistoSp diagnosed with BP demonstrated a probability of 0.69 of agreeing with HistoDx, while BIP HistoSp had the lowest likelihood (0.30) of agreeing with HistoDx. 

## 4. Discussion

Pulmonary diseases continue to be a major issue in feedyard cattle with important performance and economic impacts. Postmortem examinations in conjunction with laboratory diagnostics, such as tissue histopathology, are valuable tools to understand disease processes and the cattle affected. The goal of this study was to evaluate the frequency of three common pulmonary lesions (AIP, BP, BIP) as diagnosed grossly and by histopathology. Both diagnostic methodologies revealed similar findings in frequency of specific syndromes; however, individual diagnostic probability of agreement between the two modalities was less than expected. The potential utility of a single lung sample was evaluated by comparing diagnoses of individual samples (HistoSp) to overall HistoDx and revealed that level of diagnostic probability of agreement varied by HistoDx. 

Based on GrossDx, BP was diagnosed in 36.6% of mortalities followed by BIP in 35.8% of mortalities and AIP in 10.0% of mortalities. The HistoDx revealed 32.3% of lungs had BP, 36.0% of lungs had BIP, and 12.2% of lungs were AIP. The frequency of pulmonary lesions varies from previous studies due differences in sample size and selection of cases for specific clinical signs prior to death [22,27]. Both diagnostic modalities revealed that BP and BIP were the most common syndromes. Etiologic and risk factors for the bronchopneumonia component have been well documented in the literature with many cases involving viral and bacterial pathogens in susceptible animals [14,30,31]. This study revealed that AIP cases comprised a substantial portion of the mortalities. The prevalence observed in this study may have been higher as the study was conducted in the summer months and in a population of predominately heifers, both of which have been previously described as risk factors for AIP [18,27,32,33]. Additional cross-sectional studies at varied times of year would be helpful to more accurately describe ranges in pulmonary disease prevalence among populations and seasons.

One novel aspect of this study is recording pulmonary lesions with both bronchopneumonia and interstitial components as BIP cases. The BIP syndrome has been reported [26,27], but little documentation on the overall frequency compared to other pulmonary disease processes is available. Both GrossDx and HistoDx revealed that BIP occurs frequently in deceased feedyard cattle. One potential reason BIP was identified more frequently was the sampling method used for histopathology. Histopathological samples are often selected from diseased areas; however, in this study all cattle were sampled in the four pulmonary quadrants regardless of gross lesions or diagnosis. Interstitial pneumonic changes were identified in multiple cases, but these may have been present in other cases yet undiagnosed due histopathological focus in other areas. The pathogenesis of BIP is not well described and there have been few studies published evaluating the etiology and demographics of cattle diagnosed with BIP lesions. Cases with BIP could result from an inciting BP case followed by interstitial patterns developing from chronic lung disease or the pathogenesis may be more like the development of AIP in cases recovered from BP. Classifying BIP as a unique diagnosis can provide further insight and consistency when monitoring changes in pulmonary disease prevalence within populations. 

Despite similarities in the overall frequency of syndromes between GrossDx and HistoDx, the two diagnostic modalities only tended to be associated when comparing individual cases. The GrossDx is the most frequent form of diagnosis in deceased cattle. Histopathology is considered the “gold standard” for diagnosis of AIP lesions [28,31] and in this study HistoDx of AIP agreed with only 3 GrossDx AIP, yet 10 of the HistoDx AIP were considered BIP by GrossDx. Thus, the GrossDx should be considered a presumptive diagnosis when trying to identify AIP or BIP with HistoDx used as the final diagnosis. Previous research has reported similar values when comparing gross and histopathological diagnosis agreement [27]. In these 10 cases it is possible the histopathological sample did not contain the BP components observed grossly. The majority of cases diagnosed with BIP agreed between the two modalities and this was a frequent diagnosis. Cattle with GrossDx of BP were often called BP or BIP by HistoDx which is less surprising as the HistoDx may have observed interstitial pulmonary changes unable to be viewed grossly. This finding may indicate that acute interstitial pulmonary changes with BP cases are more common than previously reported. The low probability of agreement between GrossDx and HistoDx may be restricted to the small samples taken for laboratory evaluation. 

Clinical diagnoses are often determined solely on gross evaluation with no additional diagnostics, such as histopathology. As this study collected multiple histopathological samples per case, one of the objectives was to evaluate if a single sample would be representative of the histopathological diagnosis and if the location of the sample was important. The statistical model indicated that sample location was not significantly associated with the likelihood of a HistoSp representing HistoDx and there was no significant interaction between sample location and HistoSp. The HistoSp diagnosis was associated with HistoDx, and the level of agreement between probabilities varied by HistoDx. Cases with BIP HistoDx had the lowest probability of agreement indicating that a single HistoSp may not provide the BIP diagnosis. Based on GrossDx, investigators often observed pulmonary areas with distinctly different disease process in BIP cases; therefore, a single sample may not be representative of the entire pulmonary area. This agrees with the current etiology that acknowledges BIP cases to have BP lesions in the cranioventral lung lobes and AIP in the caudodorsal lung lobes [27]. Cases with BP HistoDx had higher probability of agreement (0.71) based on an individual sample, and disagreement may have been due to more localized lesions. If samples were taken from grossly diseased areas our expectation would be higher probability agreement of a single HistoSp with HistoDx. The probability of agreement of HistoDx AIP with a single sample was high (0.91) indicating a diffuse disease process in AIP cases and a single histopathological sample would be diagnostic in most instances. These findings indicate that pulmonary disease is common in feedlot cattle and the specific diagnosis varies by the modality utilized to evaluate pulmonary lesions.

Limitations of this study include that samples were collected based on availability of deceased animals during a limited period of time in 6 different feedyards. The estimated pulmonary disease prevalence may have differed based on season or a varied population of feeder cattle.

## 5. Conclusions

Results from this study indicated that pulmonary lesions are frequent in deceased feedlot cattle with BP and BIP representing the two most common syndromes. Refining mortality diagnostics and creating more specific descriptions of pulmonary lesions at death can facilitate better understanding of pathologic processes and appropriate therapeutic interventions. The BIP diagnosis has been infrequently acknowledged as a specific pulmonary lesion, but an improved understanding of this syndrome may play a role in identifying improved prevention and control measures. The frequency of pulmonary syndrome diagnosis was similar among GrossDx and HistoDx; however, individual case diagnoses were not statistically associated between the two diagnostic modalities. Thus, the GrossDx should be considered a presumptive diagnosis when trying to identify AIP or BIP with HistoDx used as the final diagnosis. This study collected 4 histopathological lung samples in a subset of cases and in this subset a single sample was more likely to agree with HistoDx when the HistoDx was AIP or BP compared to BIP. The three major pulmonary syndromes evaluated were common in mortalities and diagnosed with both diagnostic modalities. Refined mortality diagnostics are necessary to improve understanding of pulmonary pathology which can be valuable for evaluating and adjusting therapeutic interventions.

## Figures and Tables

**Figure 1 vetsci-10-00228-f001:**
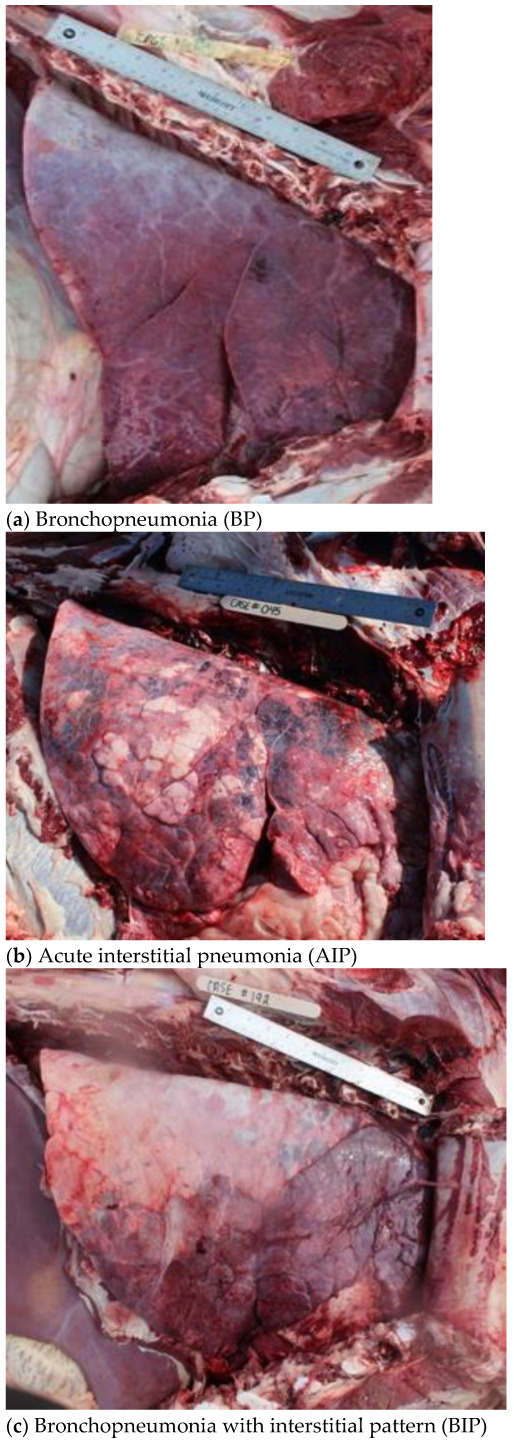
Right-sided view of gross lesion characteristics representing the three main gross pulmonary pathological diagnoses in this study: (**a**) bronchopneumonia (BP), (**b**) acute interstitial pneumonia (AIP), and (**c**) bronchopneumonia with an interstitial pneumonia (BIP).

**Figure 2 vetsci-10-00228-f002:**
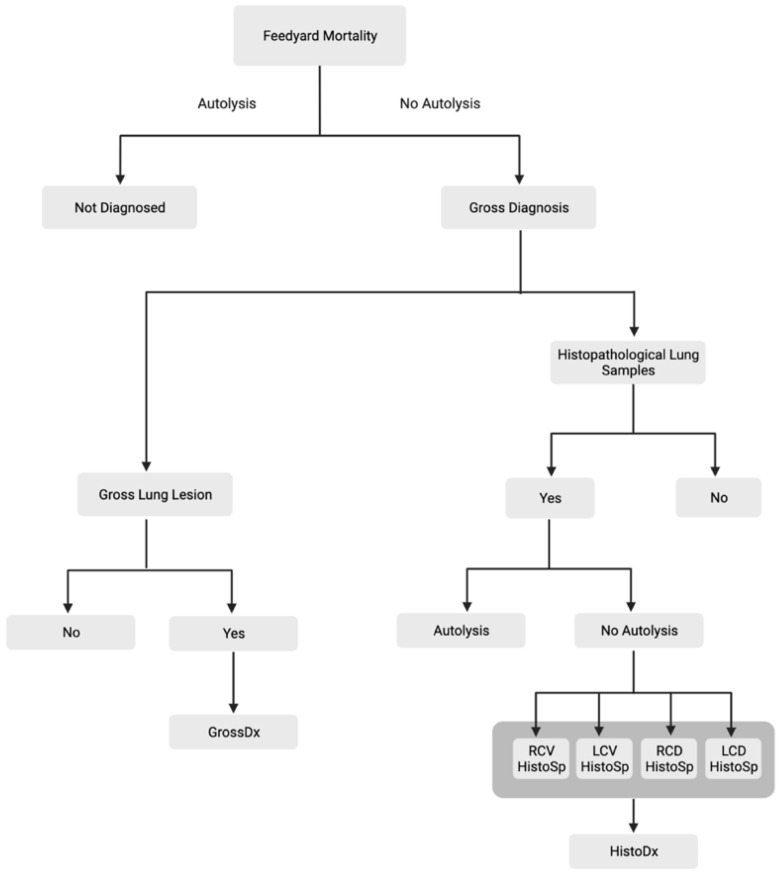
Case inclusion strategy to determine feedyard mortalities evaluated for pulmonary lesions by either gross diagnosis (GrossDx) or histopathological diagnosis (HistoDx). Four lung samples were taken from a subset of cases for histopathology. Samples were acquired from the right cranioventral (RCV), left cranioventral (LCV), right caudodorsal (RCD), and left caudodorsal (LCD) lung lobes.

**Figure 3 vetsci-10-00228-f003:**
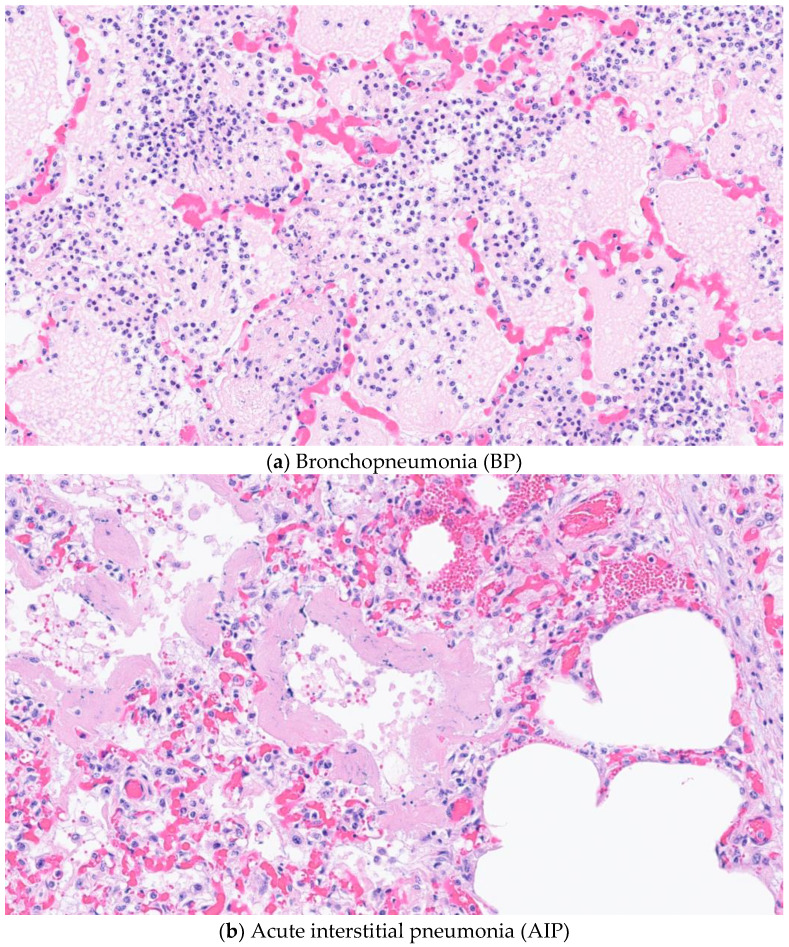
Photomicrograph examples from individual cases diagnosed with bronchopneumonia (BP) (**a**); acute interstitial pneumonia (AIP) (**b**); bronchopneumonia with interstitial pattern (BIP) (**c**).

**Figure 4 vetsci-10-00228-f004:**
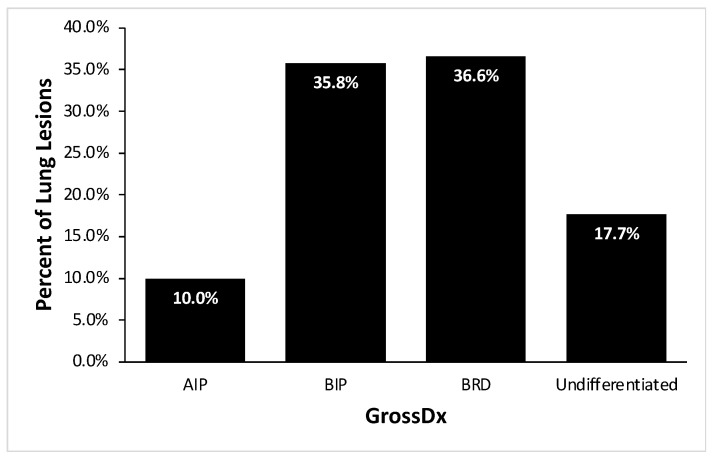
Frequency of pulmonary lesions based on gross diagnosis (GrossDx) for individual cattle (*n* = 402) diagnosed with acute interstitial pneumonia (AIP), bronchopneumonia with interstitial pneumonia (BIP), bronchopneumonia (BP), or Undifferentiated (including healthy).

**Figure 5 vetsci-10-00228-f005:**
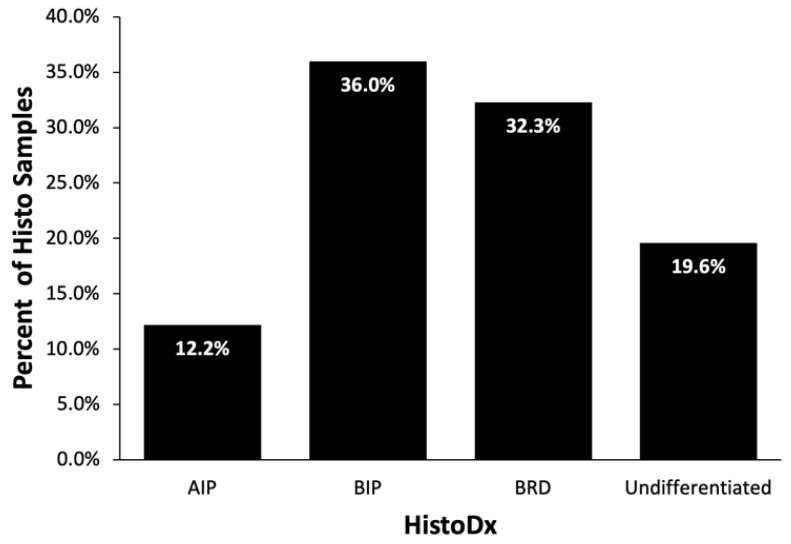
Frequency of pulmonary lesions based on histopathological diagnosis (HistoDx) for individual cattle (*n* = 189) diagnosed with acute interstitial pneumonia (AIP), bronchopneumonia with interstitial pneumonia (BIP), bronchopneumonia (BP), or Undifferentiated (including healthy).

**Figure 6 vetsci-10-00228-f006:**
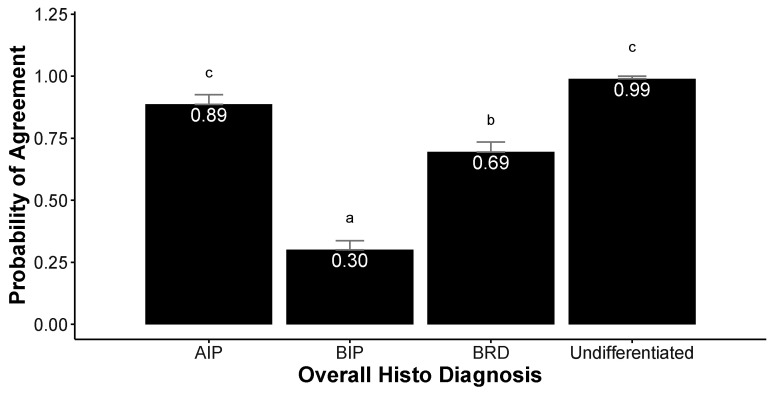
Probability of agreement between histopathological sample (HistoSp) diagnosis with overall case diagnosis (HistoDx) from feedyard necropsies on 172 cases. Results generated from generalized linear model with random effects for individual feedyard (*n* = 6) and case number (4 HistoSp per case). Differing superscript letters illustrate statistical differences (p < 0.05) in probability of agreement among HistoDx.

**Table 1 vetsci-10-00228-t001:** Descriptive table of cattle demographics for mortalities evaluated in this study.

Category	Count	Percent Total
Sex		
Heifer	279	69.40%
Steer	123	30.60%
Origin/Breed Type		
Native	374	93.00%
Beef × Dairy	24	6.00%
Holstein	3	0.70%
Mexican	1	0.20%
Arrival Weight (kg)		
181	3	0.70%
227	42	10.40%
272	81	20.10%
318	152	37.80%
363	96	23.90%
408	17	4.20%
454	6	1.50%
Days on Feed		
0–50	114	28.40%
51–100	107	26.60%
101–150	122	30.30%
151–200	51	12.70%
>200	5	1.20%

**Table 2 vetsci-10-00228-t002:** Descriptive table of gross necropsies performed at each feedyard (*n* = 417). Mortalities with gross respiratory lesions (*n* = 359) and allocation of histopathological lung samples (*n* = 318).

Week		1	2	3	4	5	6	7	8	9	Total
Feedyard 1	Total	0	5	17	22	12	7	14	12	25	114
	Gross Dx	0	5	9	22	10	7	13	12	24	102
	HistoDx	0	5	10	9	11	7	10	11	10	73
Feedyard 2	Total	1	5	6	11	10	3	2	2	6	46
	GrossDx	1	1	5	10	8	3	2	0	5	35
	HistoDx	1	2	5	10	7	2	1	2	5	35
Feedyard 3	Total	1	11	8	9	5	4	13	10	10	71
	GrossDx	1	9	6	7	5	3	10	9	9	59
	HistoDx	0	9	6	7	3	4	9	9	10	57
Feedyard 4	Total	2	6	4	5	5	2	1	2	2	29
	GrossDx	2	5	4	3	5	2	1	2	2	26
	HistoDx	2	5	4	4	4	1	1	1	2	24
Feedyard 5	Total	6	14	12	12	6	4	8	9	5	76
	GrossDx	6	12	10	8	5	4	5	7	5	62
	HistoDx	5	9	9	9	6	4	8	8	4	62
Feedyard 6	Total	3	3	13	7	14	9	11	12	9	81
	GrossDx	3	3	10	9	13	8	9	11	9	75
	HistoDx	3	3	9	7	9	8	10	9	9	67

**Table 3 vetsci-10-00228-t003:** Individual case classification based on HistoDx or GrossDx in cases (*n* = 172) diagnosed with each methodology as either acute interstitial pneumonia (AIP), bronchopneumonia with interstitial pneumonia (BIP), bronchopneumonia (BP), or Undifferentiated (including healthy). Shaded cells represent cases where HistoDx and GrossDx showed agreement.

	Histopathological Diagnoses			
Gross Diagnosis	AIP	BIP	BP	Undifferentiated
AIP	3	8	1	1
BIP	10	44	32	3
BP	6	13	21	18
Undifferentiated	3	2	5	2

## Data Availability

Data utilized for this research were from cooperating entities and are not available publicly due to confidentiality and anonymity agreements.

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
