# Peer review of "Determining Frequency of Common Pulmonary Gross and Histopathological Findings in Feedyard Fatalities"

_vetsci, 2023, doi:10.3390/vetsci10030228_

Round 1
Reviewer 1 Report
I think the article should be completed by microbiological or molecular tests to understand the real etiology of the lung pathologies detected. The authors carried out autopsies on animals that died in the summer so it is interesting to understand the cause of those pathologies.Author Response
Reviewer 1 Report:
I think the article should be completed by microbiological or molecular tests to understand the real etiology of the lung pathologies detected. The authors carried out autopsies on animals that died in the summer so it is interesting to understand the cause of those pathologies.
Author’s Response:
AU: Thank you for taking the time to review our paper. We agree etiology of the lung pathologies would be beneficial information, however those diagnostics were beyond the scope of this study and its design. Our objective for this study was to evaluate gross and histopathological lesions, but it would valuable if subsequent studies could incorporate further information on etiological diagnoses.
Reviewer 2 Report
The stated goal of the study was to improve understanding pulmonary pathology in feedlot cattle for evaluating and adjusting therapeutic interventions. As designed, the study adds little to our knowledge of bovine respiratory disease in feed yard cattle. The lack of cultures, IHC, or ISH to identify bacterial and viral pathogens renders the results useless. Lastly, the criteria to classify the cases ignores a large body of literature that critically evaluates gross and microscopic lung lesions that helps prioritize specific etiologic agents.
An unstated goal of the study appears to be an attempt to substitute gross pathology diagnosis for histopath diagnosis. If nothing else, the study documents both diagnostic modalities are complementary
Lines 151-164: excess detail. Replace with: Formalin-fixed tissues were processed using standard histopathological techniques.
Author Response
Reviewer 2 Report:
The stated goal of the study was to improve understanding pulmonary pathology in feedlot cattle for evaluating and adjusting therapeutic interventions. As designed, the study adds little to our knowledge of bovine respiratory disease in feed yard cattle.
AU: Thank you for taking the time to review our paper and make suggestions to improve the manuscript. However, we respectfully disagree the information gained through this project does not contribute to our knowledge of pulmonary disease in cattle. This report describes differences in frequency of disease syndromes based on gross diagnosis and compared to histopathological diagnosis. Furthermore, BIP has been infrequently reported as its own category of pulmonary disease. This study demonstrated the frequency of this specific pulmonary disease, which can promote further research to pursue the specific disease process and etiologies.
The lack of cultures, IHC, or ISH to identify bacterial and viral pathogens renders the results useless.
AU: We acknowledge the identification of bacterial and/or viral pathogens is valuable to describe feedlot lung disease; however, determining the specific etiologic agents was beyond the scope of this study. The study design was to descriptively analyze feedlot moralities through systematic necropsies (which are not routinely performed) and the frequency of occurrence.
Lastly, the criteria to classify the cases ignores a large body of literature that critically evaluates gross and microscopic lung lesions that helps prioritize specific etiologic agents.
AU: We have tried to incorporate previous literature relative to gross and histopathological diagnosis.
An unstated goal of the study appears to be an attempt to substitute gross pathology diagnosis for histopath diagnosis. If nothing else, the study documents both diagnostic modalities are complementary
AU: As stated in the objective the goal was to evaluate both gross and histopathological findings. For some pulmonary disease (e.g. AIP) histopathology would be considered the gold standard of diagnosis. We agree the methods are complimentary and we feel this manuscript illustrates the relationship between the two modalities.
Lines 151-164: excess detail. Replace with: Formalin-fixed tissues were processed using standard histopathological techniques.
AU: We have shortened this section.
Reviewer 3 Report
Comments / suggestions are attached as a separate document.

Author Response
Reviewer 3 Report:
Schmidt et al. described the relative frequency of suspected (via gross postmortem exam) and confirmed (via histology) cases of bronchopneumonia, AIP, and the recently proposed morphologic diagnosis “BIP” among calves from six feedlots in the summer of 2022. The goal of the study was two-fold: 1) to determine the relative frequency of lung lesions that characterize the three syndromes and 2) to determine the agreement between gross and histologic diagnoses of the cases. The study found moderate agreement between the gross and histologic characterizations that approached significance, which highlights the importance of both gross necropsy and histology in disease investigations of BRD.
AU: thank you for your review of the paper; we agree this paper highlights the importance of utilizing both gross and histological analyses.
The frequencies reported here are fairly similar to original inquiries using the new BIP designation in Canada and add to the body of knowledge surrounding BRD as a whole, and in particular to the newly proposed morphologic diagnosis of BIP.
In my opinion, the syndrome characterized as BIP by Haydock et al. and referenced in this paper is in some ways giving a new name to something that pathologists have historically seen. The impact of defining this as a separate syndrome remains to be seen, although differences in epidemiology between BIP, AIP, and bronchopneumonia previously reported (time of year, heifers vs steers, DOF, etc..) are evidence for its own designation distinct from bronchointerstitial pneumonia, or any other combination thereof. Data such as that provided in this paper will be important to determine the validity of creating this new syndrome name moving forward.
AU: We agree this BIP syndrome merits further investigation and several reports have described the process in similar. We think this may indeed be providing a name for a syndrome previously observed; however, it is uncommon to see feedlot deaths recorded in normal record systems with this syndrome. Our goal with this research is to highlight frequency to encourage other researcher for further investigation of epidemiological, etiological, and other descriptions of this syndrome.
Overall, the study design, statistical analysis and interpretation, and flow of the manuscript text is well done. There are some discrepancies within the paper, and between this paper and other papers regarding nomenclature, phrases used, and definitions that I think are at times confusing as a reader. Below are my attempts to highlight these points and offer suggestions to improve the manuscript in that regard.
AU: We appreciate your comments and have tried to make changes to the manuscript to improve clarity. We placed specific responses to your suggestions below.
General comments/suggestions without specific line numbers:
- Both AIP and BIP, require the presence of type II pneumocyte hyperplasia and/or hyaline membranes lining alveoli by definition. Therefore, Table 1 of Haydock (2023) and Figure 4 of this paper, are claiming that 63.4% and 48.2% of lungs displayed one or both of these characteristics in the respective studies. As a former feedlot veterinarian and current diagnostic pathologist, these numbers seem exceptionally high. I think these apparently high percentages are likely the result of either one or both of the following:
AU: We agree the amount of BIP observed was higher than we expected also. One unique aspect of this study was that pulmonary tissue was assessed in 4 locations regardless of lesion location or gross diagnosis. As you highlighted below in b. this may be one of the reason we observed some differences.
- Potential individual differences among pathologists in what qualifies as a hyaline membrane. We can’t avoid the inherent subjectivity of histology, but I would suggest a brief mention of what was used to characterize hyaline membranes in the M & M. Although not crucial in my opinion, a figure of photomicrograph examples of each of the 3 syndromes may be beneficial as well.
AU: We spoke to the pathologist that read the slides and have added to the text directly from the pathologist: “Hyaline membrane are characterized by eosinophilic lamellations of polymerized fibrin and necrotic debris that replace lost type I pneumocytes of the alveolar interstitium. “ We have also added figure with photomicrograph examples of each of the three syndromes.
- In diagnostic cases (and past research studies), the obviously consolidated and affected cranioventral portion of lung are often the only parts examined. It is possible that high numbers of hyaline membranes and/or type II pneumocyte hyperplasia in other areas of diseased lungs have just gone unnoticed since we don’t routinely look there. If the authors believe the pathologist in this case is correctly classifying hyaline membranes relative to other pathologists (which I have no reason not to assume is true), then I think bringing up the point about historically not looking elsewhere in the manuscript would be a good addition.
AU: We agree and this is a great point; we have added to the discussion.
- Somewhat related to #1, but since BIP and AIP require specific histologic characteristics to make a definitive diagnosis, any attempts to make the diagnosis on gross lesions/characteristics alone are at best, presumptive (i.e. an educated guess). I think within the paper it is entirely appropriate to describe gross characteristics observed in cases of BIP and AIP (that were later confirmed via histology). But, assigning a “diagnosis” of AIP or 2 BIP on gross alone is technically incorrect. I realize this type of wording is commonly used in the literature (including by Haydock et al.) and what is really meant is “a guess”, but this wording seems inappropriate given the requirement of histology to meet the defined diagnostic criteria. While I don’t view it as crucial to the manuscript, highlighting the fact that these require histology by definition and therefore any gross diagnoses are guesses would provide some clarity to the reader. Another potential suggestion is to replace “gross diagnosis” with presumptive or suspected gross diagnosis throughout the text.
AU: We agree; this is a good point. the results show that we didn’t always have good agreement when determining cases to be BIP/AIP grossly. Histopathology is certainly the gold standard for the interstitial pnueumonia; however, we used the gross diagnosis terminology because as you stated this is standard in the literature. We have added to the discussion to capture this point.
Below specific points and suggested changes are highlighted by line number.
AU: Thank you for taking the time to review our paper and the thorough suggestions to improve the manuscript. Below we have addressed your suggestions:
Line 18: “A diagnosis was determined”. I suggest calling this a presumptive gross diagnosis as mentioned in the above paragraph.
AU: corrected as suggested
Line 20: diagnostics > diagnostic
AU: corrected as suggested
Line 23: “bronchopneumonia with an interstitial pneumonia” (line 15) and “bronchopneumonia with an interstitial pattern” (numerous lines) are used interchangeably in the text. On line 23 (and throughout the text) I suggest using only “bronchopneumonia with interstitial pneumonia” as defined by Haydock when describing the BIP syndrome since that definition is the basis for this paper in large part.
AU: Thank you for pointing out the variability of terms. The following lines have been changed to all read “bronchopneumonia with an interstitial pneumonia”: Line 26, 32, 42, 47, 91, 100, 133, Figure 1, 287, Figure 4 description, and Table 3 description.
Line 64: “Mortality diagnosis” is a confusing term to me. I would suggest either postmortem etiologic diagnosis or cause of death depending upon the precise point you’re trying to make here.
AU: The sentence has been changed to read: Line 71 “Cause of death is often recorded as a…”
Line 67: Pathology is the study of disease. I would suggest saying “concurrent non-pulmonary disease” or “concurrent non-pulmonary lesions”
AU: The sentence has been changed to read: Line 74 “or concurrent pathological lesions can provide…”
Line 68: “classification of mortalities” is confusing to me again like line 64. I suggest replacing with “improved classification regarding cause of death” or something similar.
AU: Line 75 was modified to: “Improved classification regarding cause of death seen throughout…”
Lines 73/74: This is improper use of the acronym BRD in my opinion. I suggest using BRD only when referring to the umbrella term “bovine respiratory disease” throughout the paper. BRD most accurately is used when describing a clinical diagnosis encompassing respiratory disease from any of the potential causes (bacterial/viral/toxic/combination). Bronchopneumonia is technically a morphologic diagnosis (not a syndrome as mentioned here on line 73), and not the only morphologic diagnosis that would qualify as BRD since interstitial pneumonia or bronchointerstitial pneumonia would also be correctly placed under the BRD umbrella. I think there are many times in the text when you say “BRD” where you are actually referring to bronchopneumonia. Particularly on the following lines: 73-74, 87, 92, 184, 186, 187, 188, 236, 253, 298, 304, 306, 309, 325, 326, 338, 339, 355, 357, 371
AU: this is a good point and to be consistent with other literature we have modified the manuscript to change these to BP instead of BRD.
Line 80: “pathologic lesions” is a redundant term since lesions are by definition pathologic. Suggest just saying “lesions”.
AU: The sentence has been edited to read: Line 87 “gross lesions of BRD…”
Line 81: Citing Blood’s original 1962 (Can Vet J) characterization of AIP is warranted in my opinion here when introducing AIP. The original characterization was before the idea that there are two types of AIP (pasture vs feedlot), but this still qualifies as the original paper defining the diagnosis I believe.
AU: this is a good suggestion and we have added this citation
Lines 84/85: Here you’re introducing the term and abbreviation for BIP, but you’re citing Zhang et al. I think you likely meant to cite the Haydock paper. Also, in the Zhang paper they do use the abbreviation BIP, but for “bronchointerstitial pneumonia” and not for “bronchopneumonia with interstitial pneumonia” like you’re doing for this paper. In the Zhang paper they use the abbreviation BP + BIP to describe what Haydock (and you) refer to as BIP. I suggest citing Haydock instead of Zhang, replacing pattern with pneumonia, and replacing BRD with bronchopneumonia for this sentence.
AU: Corrected as suggested.
Line 91: I would suggest using BP for bronchopneumonia abbreviation. I also suggest pattern > pneumonia
AU: Changed as suggested.
Line 116: pathological pulmonary lesions is redundant. Suggest “gross lung lesions” or something similar.
AU: The sentence has been edited to read: Line 125 “determining the gross lesions.”
Line 120: I think you mean interlobular edema here?
AU: We debated this terminology prior to submission. To stay consistent with Haydock et al. the term was modified to say “interlobular edema” (Line 132).
Line 124: “diagnosed with a respiratory lesion” suggest “diagnosed as”
AU: The sentence was edited to read: Line 136 “interstitial pneumonia were diagnosed as BIP.”
Line 124/125: pathological lesions is redundant. Suggest just saying lesions.
AU: Lines 128 & 136 were edited to read: “gross pulmonary lesions…”.
Line 144: “11 samples”, suggest “11 cases”.
AU: The sentence was edited to read: Line 168 “11 cases were taken…”
Line 151 – 163: This paragraph seems superfluous. I think it would be sufficient to summarize as: “Fixed tissues were processed routinely for histology and 5 µm sections stained with H&E. Slides were scanned using…”. I don’t see anything in the protocol out of the ordinary that would warrant so much explanation.
AU: We have shortened this section as suggested.
167-168: As mentioned above, I think here would be good place to either cite or define what you’re using for the specific histopath criteria.
AU: this has been added to the text.
168-172: I understand what you’re saying here, but I think it could be cleared up some. A characteristic finding of AIP is traditionally an absence of inflammation (See Blood’s original paper) so mentioning interstitial expansion by mononuclear cells here implies that was included as required criteria. I suggest mentioning that by definition all AIP/BIP cases had either hyaline membranes and/or type II pneumocyte hyperplasia. Then mentioning that septal inflammation, septal necrosis, etc.. were also sometimes seen in these cases to differing degrees as well. I would suggest omit the mention of being nonspecific for type II pneumocyte hyperplasia and bronchiolitis obliterans because technically septal inflammation and septal necrosis are also nonspecific findings.
AU: changed as suggested
Line 174: I suggest omitting “or a cause could not be elucidated” since you’re not technically assigning a cause to any of these cases. You are solely assigning a morphologic diagnosis (not etiologic diagnosis).
AU: The following edits were made to line 197: “For the purpose of this study, histopathological lung samples that did not meet the above criteria were categorized as Undifferentiated.”
Line 186: pathology is the study of disease. Suggest “but no lesions consistent with AIP” or something similar.
AU: The followed edits were made to line 208: “ but no lesions consistent with AIP were noted in BRD cases…”.
Table 1: The alignment of some columns is off.
AU: corrected as suggested
Line 233: pathology = study of disease. Suggest “lung lesions”
AU: corrected as suggested
Line 311-312: Suggest replacing “significant” with “substantial” unless you are able to assign a p value.
AU: changed as suggested
Line 312 – 315: This is an excellent point and something I thought of immediately when seeing the study was done only in the summer. Determining if “BIP” is as prevalent in non-summer months as it was in this study will help validate whether or not this new name is a necessary specific designation or not.
AU: Thank you for your comment. We agree that more research is needed to gain knowledge about bronchopneumonia with an interstitial pneumonia and the risk factors involved. We would like to expand our sample collection period to different seasons and geographic regions in future research.
Line 318 – 328: This paragraph would be a good place to point out (assuming authors agree) that histologic characteristics such as hyaline membranes or type II pneumocyte hyperplasia may be more prevalent than traditionally thought and that historically we don’t see them simply because we don’t look in the CD part of the lung in a regular case of suppurative bronchopneumonia in a calf in a feedlot setting.
AU: we agree and have added sentences to express this thought.
Line 333: My version does not have a reference 33
AU: Thank you for pointing this out. There should not have been a #33 reference, the in-text citation was deleted.
Line 340 – 342: This is along the lines of what I’ve eluded to above. I think pointing out it’s potentially more common than previously reported is because we very seldom look outside of the CV lung in diagnostic cases in particular, but also in many research studies.
AU: we agree and have added to the discussion
Line 344-345: This sentence I think is confusing. Histo samples provide aid to a clinical diagnosis, but not a gross diagnosis. Gross diagnoses are by definition gross only. I suggest “Clinical diagnoses often rely solely on gross interpretation” or something similar.
AU: Line 415-416 modifications: “Clinical diagnoses are often determined solely on gross evaluation with no additional diagnostics, such as histopathology. “
Line 371 – 372: Yes, it’s been infrequently reported, but that’s because it was only recently proposed as a novel morphologic diagnosis. I think the way this sentence is structured it implies it’s an emerging disease syndrome.
AU: The sentences has been edited to read: Line 445 “The BIP diagnosis has been infrequently acknowledged as a specific pulmonary lesion, but…”.
Reviewer 4 Report
The authors of the manuscript described the utilisation of necropsy and histopathology to determine the frequency of three different pulmonary pathologies occurring in feedlot cattle during the summer months.
The results of 417 necropsies and histopathological examination of 189 cases were analysed using descriptive statistics. The authors compared the gross pathology diagnoses with histopathological ones revealing that there was a trend (p=0.06) that both methods resulted in the same diagnosis.
As the authors indicated, each recognised pathological pattern (bronchopneumonia, acute interstitial pneumonia and broncho-interstitial pneumonia) is associated with different aetiology in pulmonary disease in cattle therefore correct diagnosis determines the prevention and/or successful treatment.
While the manuscript describes detailed analyses followed by appropriate discussion and relevant references, there are certain areas requiring improvement :
It is difficult to conclude in what way the paper provides an advancement to the current knowledge. The authors suggest that the BIP diagnosis is a novel aspect of the study (line 370), however, there are publications that already established such pathological pattern in bovine pulmonary disease, as the authors also noted.
The paper lacks a clear conclusion- how the results of the presented analysis can contribute to making decisions aiming at solving problems of cattle mortalities due to pulmonary diseases?
1. Page 3 2.1. Experimental design.
Where the cattle mortalities occurred? Information indicating the country/region of the disease occurrence might help distinguish the present work from other reports, considering that recently similar papers have been published (e.g. positions 26 and 27 in the References section - both papers by Haydock et.al.). It might also be important in terms of comparative analyses of bovine respiratory diseases occurring in different geographic regions.
2. Page 4 Figure 2. Explain the abbreviations RCV, LCV, RCD, and LCD in the figure description.
3. Page 4 2.3 Histopathology: the sections 1 cm x1cm seem to be quite small for histopathology – given, that the GrossDx were not always overlapping with HistoDx (especially in BIP cases) did you take into consideration that the tissue size could influence the HistoSp and HistoDx diagnosis?
4. Line 153: Please replace “Excelsior” which is a trade-name, with “tissue processor”
5. Page 7 / 3.1 Gross Results
Lines 233-237 The description of the results is confusing. In Line 235 the sentence says “Of the 402 cases that were diagnosed with a lung lesion, 36.6% (147/402) …" which is not in accordance with the former sentences (lines 233-234): “Of the 402 cattle with gross necropsy results, 359 cases showed gross lung pathology (89%). The remaining forty-three cases were not evaluated due to the absence of a lung lesion”
What is the correct number of analysed cases with lung pathology?
6 . Page 8. 3.3 GrossDx versus HistoDx
The presentation of probability values is inconsistent. For example on Page 8. in paragraph 3.3 GrossDx versus HistoDx there are symbols such as P=0.49 used for the probability of agreement, but in paragraph 3.4 line 285 these symbols are not used. There is also P used as a p-value e.g P=0.06 which describes statistical significance. In that form, these statistical figures are difficult to follow in the text.
7. Conclusions.
Line 379 – what do the authors mean by “refined mortality diagnositcs”? Which results lead to this conclusion?
Author Response
Reviewer 4 Report:
The authors of the manuscript described the utilisation of necropsy and histopathology to determine the frequency of three different pulmonary pathologies occurring in feedlot cattle during the summer months.
The results of 417 necropsies and histopathological examination of 189 cases were analysed using descriptive statistics. The authors compared the gross pathology diagnoses with histopathological ones revealing that there was a trend (p=0.06) that both methods resulted in the same diagnosis.
As the authors indicated, each recognised pathological pattern (bronchopneumonia, acute interstitial pneumonia and broncho-interstitial pneumonia) is associated with different aetiology in pulmonary disease in cattle therefore correct diagnosis determines the prevention and/or successful treatment.
While the manuscript describes detailed analyses followed by appropriate discussion and relevant references, there are certain areas requiring improvement :
It is difficult to conclude in what way the paper provides an advancement to the current knowledge. The authors suggest that the BIP diagnosis is a novel aspect of the study (line 370), however, there are publications that already established such pathological pattern in bovine pulmonary disease, as the authors also noted.
AU: We appreciate your comments relative to the paper and have tried to make revisions to address the points that you identified. The objective of this study was to evaluate frequency of pulmonary pathologies and evaluate gross and histopathological lesions. Through this process we described BIP as a frequent syndrome which has only been recently reported. We feel this work is additive to the literature relative to the description of this syndrome.
The paper lacks a clear conclusion- how the results of the presented analysis can contribute to making decisions aiming at solving problems of cattle mortalities due to pulmonary diseases?
AU: One important conclusion of this paper is how the cause of death is recorded in feedyard mortalities. Most pulmonary disease is recorded as bronchopneumonia or AIP; however, if BIP represents a separate category this should be reported so it can be investigated appropriately.
- Page 3 2.1. Experimental design.
Where the cattle mortalities occurred? Information indicating the country/region of the disease occurrence might help distinguish the present work from other reports, considering that recently similar papers have been published (e.g. positions 26 and 27 in the References section - both papers by Haydock et.al.). It might also be important in terms of comparative analyses of bovine respiratory diseases occurring in different geographic regions.
AU: Due to feedyard confidentiality, we can not to list the specific feedyards. However I modified to wording to state: Line 106 & 107 “Feedlot cattle were necropsied at six feedyards from June 1, 2022 to July 29, 2022, located in the high plains region of the United States”. Thus the study geographic region differed from the Haydock papers and we found similar findings.
- Page 4 Figure 2. Explain the abbreviations RCV, LCV, RCD, and LCD in the figure description.
AU: The following sentences were added to the Figure 2 description. Line 146 -148 “ Four lung samples were taken from a subset of cases for histopathology. Samples were acquired from the right cranioventral (RCV), left cranioventral (LCV), right caudodorsal (RCD), and left caudodorsal (LCD) lung lobes.”
- Page 4 2.3 Histopathology: the sections 1 cm x1cm seem to be quite small for histopathology – given, that the GrossDx were not always overlapping with HistoDx (especially in BIP cases) did you take into consideration that the tissue size could influence the HistoSp and HistoDx diagnosis?
AU: We chose 1cm X 1cm lung samples to match the recommended 1:10 tissue to formalin ratio. This size was recommended by the pathologist in during the planning stage of the project. However, in the discussion Line 380- 382, 386-387, and 398-399 we acknowledge that our small sample could be a limiting factor in correlation between the two disease processes.
- Line 153: Please replace “Excelsior” which is a trade-name, with “tissue processor”
AU: Thank you for the suggestion. The sentence has been modified to: Line 163 “ The cassettes with fixed tissues were placed in a tissue processor and processed according to manufacturer’s protocol.”
- Page 7 / 3.1 Gross Results
Lines 233-237 The description of the results is confusing. In Line 235 the sentence says “Of the 402 cases that were diagnosed with a lung lesion, 36.6% (147/402) …" which is not in accordance with the former sentences (lines 233-234): “Of the 402 cattle with gross necropsy results, 359 cases showed gross lung pathology (89%). The remaining forty-three cases were not evaluated due to the absence of a lung lesion”
What is the correct number of analysed cases with lung pathology?
AU: Thank you for your suggestion. All 402 cases were considered in the analysis and the 43 cases that did not have gross lung pathology were categorized into the Undifferentiated category. The following modification was made to Line 165 “Of the 402 cases that were grossly diagnosed, 36.6% (147/402)…”
6 . Page 8. 3.3 GrossDx versus HistoDx
The presentation of probability values is inconsistent. For example on Page 8. in paragraph 3.3 GrossDx versus HistoDx there are symbols such as P=0.49 used for the probability of agreement, but in paragraph 3.4 line 285 these symbols are not used. There is also P used as a p-value e.g P=0.06 which describes statistical significance. In that form, these statistical figures are difficult to follow in the text.
AU: we have changed these to be listed as ‘p-value=’ to alleviate any potential confusion.
- Conclusions.
Line 379 – what do the authors mean by “refined mortality diagnositcs”? Which results lead to this conclusion?
AU: Currently, many feedyard mortalities that are necropsied and diagnosed are broad and provide very little description about lung pathology. We address this issue in the introduction (Line 90- 91). Acknowledging BIP as it’s own separate entity and not classifying it as an AIP or BRD lesion can guide future knowledge about the specific etiology of this disease. This concept is also addressed in the discussion on Line 374. By refining our mortality diagnostics and being more specific about lung lesions at time of death, we believe we can better understand the pathological processes occurring and the appropriate therapeutic interventions.
Round 2
Reviewer 4 Report
Thank you for the replies. The statement "The objective of this study was to evaluate frequency of pulmonary pathologies and evaluate gross and histopathological lesions. Through this process we described BIP as a frequent syndrome which has only been recently reported." is an important summary of the study (particularly emphasizing, that BIP is a relatively new diagnostic term) and it should be included in an abstract. Lines 365-366: "Thus, the GrossDx should be considered a presumptive diagnosis when trying to identify AIP or BIP with HistoDx used as the final diagnosis." - this is one of the conclusions of the study and should be noted in the Conclusions section, given, that the comparison between GrossDx and HistoDx was an important part of the study. Re: reply to the question 7: Thank you for the answer. I think the last sentence should follow the second one in the "Conclusions" - it is easier to follow your thought process this way.
Author Response
Reviewer 4 report:
Thank you for the replies. The statement "The objective of this study was to evaluate frequency of pulmonary pathologies and evaluate gross and histopathological lesions. Through this process we described BIP as a frequent syndrome which has only been recently reported." is an important summary of the study (particularly emphasizing, that BIP is a relatively new diagnostic term) and it should be included in an abstract.
AU: thank you for your constructive comments to help review the manuscript. We have the objective in the abstract and added the second sentence emphasizing the identification of frequent BIP.
Lines 365-366: "Thus, the GrossDx should be considered a presumptive diagnosis when trying to identify AIP or BIP with HistoDx used as the final diagnosis." - this is one of the conclusions of the study and should be noted in the Conclusions section, given, that the comparison between GrossDx and HistoDx was an important part of the study.
AU: this is a good point and we have added that sentence to the conclusions
Re: reply to the question 7: Thank you for the answer. I think the last sentence should follow the second one in the "Conclusions" - it is easier to follow your thought process this way.
AU: this is a good addition and we have added this sentence after the initial sentence in the conclusions